# Mutation Patterns in Portuguese Families with Hereditary Breast and Ovarian Cancer Syndrome

**DOI:** 10.3390/cancers14194717

**Published:** 2022-09-28

**Authors:** Rodrigo Vicente, Diogo Alpuim Costa, Marina Vitorino, Ana Duarte Mendes, Catarina Santos, Mário Fontes-Sousa

**Affiliations:** 1Medical Oncology Department, Hospital Professor Doutor Fernando Fonseca, IC19 276, 2720-276 Amadora, Portugal; 2Breast Cancer Unit, CUF Oncologia, 1998-018 Lisbon, Portugal; 3NOVA Medical School, Faculdade de Ciências Médicas, 130, 1169-056 Lisbon, Portugal

**Keywords:** hereditary breast and ovarian cancer syndrome, breast cancer, ovarian cancer, founder mutations, Portuguese families, genetic testing

## Abstract

**Simple Summary:**

The pattern of Breast Cancer Genes 1 (*BRCA1*) and 2 (*BRCA2*) mutations in Hereditary Breast Ovarian Cancer (HBOC) families varies widely among different populations. About 30% of Portuguese HBOC can be associated with inherited cancer caused by *BRCA1* or *BRCA2* mutations. Three variants were identified (c.156_157insAlu in the *BRCA2* gene and c.3331_3334del and c.2037delinsCC in the *BRCA1* gene), accounting for about 50% of all Portuguese pathogenic mutations. Characterising the mutational spectrum in specific populations allows for a more efficient and cost-saving screening approach.

**Abstract:**

Germline pathogenic variants in the Breast Cancer Genes 1 (*BRCA1*) and 2 (*BRCA2*) are responsible for Hereditary Breast and Ovarian Cancer (HBOC) syndrome. Genetic susceptibility to breast cancer accounts for 5–10% of all cases, phenotypically presenting with characteristics such as an autosomal dominant inheritance pattern, earlier age of onset, bilateral tumours, male breast cancer, and ovarian tumours, among others. *BRCA2* pathogenic variant is usually associated with other cancers such as melanoma, prostate, and pancreatic cancers. Many rearrangements of different mutations were found in both genes, with some ethnic groups having higher frequencies of specific mutations due to founder effects. Despite the heterogeneity of germline *BRCA1/BRCA2* mutations in Portuguese breast or/and ovarian cancer families, the first described founder mutation in the *BRCA2* gene (c.156_157insAlu) and two other variants in the *BRCA1* gene (c.3331_3334del and c.2037delinsCC) contribute to about 50% of all pathogenic mutations. Furthermore, the families with the *BRCA1* c.3331_3334del or the c.2037delinsCC mutations share a common haplotype, suggesting that these may also be founder mutations in the Portuguese population. Identifying specific and recurrent/founder mutations plays an important role in increasing the efficiency of genetic testing since it allows the use of more specific, cheaper and faster strategies to screen HBOC families. Therefore, this review aims to describe the mutational rearrangements of founder mutations and evaluate their impact on the genetic testing criteria for HBOC families of Portuguese ancestry.

## 1. Introduction

According to the World Health Organization, in 2020, there were 2.3 million women diagnosed with breast cancer (BC) and 685,000 deaths worldwide. It is considered the world’s most prevalent cancer and the most frequent cause of lost disability-adjusted life by women when compared to other cancer types [1]. Furthermore, ovarian cancer (OC) is one of the most common gynecologic cancers ranking eighth in terms of cancer incidence and mortality among women globally. Also known as a “silent killer”, OC has a lower prevalence than BC, but it is three times more lethal. Most women are diagnosed with advanced stage OC, exhibiting poor prognosis and consequently much less variability in mortality between low- and high-income countries [2]. Besides, a general population screening cannot be recommended [3]. 

Germline mutations in Breast Cancer Gene (*BRCA)1* and *2* account for a large proportion of hereditary breast and ovarian cancer (HBOC) families [4]. The HBOC is a known example of a genetic syndrome involving *BRCA* mutations. It can affect men and women and usually leads to BC and/or OC before age 50. Typically, it is associated with a family history of cancer, and bilateral tumours may also occur [4]. Over time, despite technical difficulties due to the large size of both genes, screening for mutation in *BRCA* in different populations was performed according to the familiar cancer patterns reported. Within well-defined populations across many countries, specific frequent mutations have been identified. The founder effects generally occur in a pool of reduced genetic heterogeneity and facilitate carrier detection and genetic counselling [5]. This was the case of c.156_157insAlu *BRCA2* rearrangement, firstly identified in a Portuguese resident in Belgium and now recognised as a founder mutation in Portuguese HBOC families [6]. This review aims to describe the mutational spectrum of Portuguese founder/recurrent mutations and reflect on their clinical implications regarding genetic testing criteria.

## 2. Genetic and Clinical Features of Hereditary Breast and Ovarian Cancer 

HBOC syndrome is defined as an autosomal dominant inherited disorder in which the risk of BC and OC is higher than normal. Globally, the hereditary predisposition cases represent 5–10% of all BC and 20–25% of OC [7]. HBOC is caused by germline mutations in *BRCA* 1 and 2, which are suppressor genes that encode proteins involved in DNA repair [4]. Those genes are responsible for repairing DNA double-strand breaks homologous recombination (HR) that provides accurate recombination using a sister chromatid as a template, allowing genomic stability [4]. Alterations of the BRCA1/2 genes may also occur through mechanisms other than germline mutations, for example, somatic mutations or epigenetic silencing in sporadic OC or BC [8].

Several other proteins interact and cooperate with BRCA1/2 in the DNA repair process and can also be key factors in cancer susceptibility, particularly in BC and OC, that present a defect in the HR system [8]. Since *BRCA1/2* mutations can have clinical and therapeutic consequences, it has been hypothesised that these alterations can also be sensitive to DNA-damaging target agents. 

In the *BRCA1*-mutated population, the risk of BC ranges from 46–87% up to age 70, while the risk of OC is 39–63%. For the *BRCA2*-mutated population, the lifetime risk of developing BC or OC is 38–84% and 16–27%, respectively [4]. The clinical characteristics of *BRCA*-associated cancers may differ from the sporadic ones. *BRCA1*-related breast tumours are frequently associated with a medullary histological pattern, higher grade and more likely to be oestrogen and progesterone receptors negative and without human epidermal growth factor receptor (HER2) overexpression. *BRCA2*-related tumours are more heterogeneous than *BRCA1*. It appears to have a predominance in positive hormone receptors BC, but around 16% of triple-negative tumours (TNBC) in *BRCA2* carriers were described [9]. The patients with *BRCA1/2* mutations have an increased risk of developing contralateral BC, around 2%, with the higher risk corresponding to the younger age of onset [4]. Also, these mutations spectrum confers a higher risk of male BC although the association is more frequently reported with the *BRCA2* variant. The cancer is usually high grade in males, with hormone receptors positive and with lymph node metastases [10,11]. 

Studies on BC survival suggest poor survival in individuals with a *BRCA* mutation, but this association has not been consistent and remains a controversial topic [12,13,14,15]. The *BRCA*-related OC (including fallopian tube and primary peritoneal cancers) are generally serous adenocarcinomas, as opposed to mucinous or borderline tumours. Ovarian cancer is more related to *BRCA1* mutations and tends to develop at an earlier age in these women [16].

Identifying these *BRCA1/2* mutations is crucial to planning the treatment and follow-up of cancer patients. In BC, international recommendations suggest a bilateral mastectomy as a primary surgical treatment in *BRCA* carriers because of their high rate of ipsilateral and contralateral BC. These patients’ systemic treatment options are increased, mostly due to the emerging poly (ADP-ribose) polymerase (PARP) inhibitors [17]. These drugs blocking PARP action and deficient *BRCA* synergistically lead to a failure in DNA repair and, consequently, to the death of tumour cells (synthetic lethality). Actually, in consequence of demonstrated benefits of PARP inhibitors, there is evidence for using these drugs in early and advanced BC settings and also in OC or fallopian tubes or primary peritoneal cancer [18,19]. 

Four PARP inhibitors are approved by U.S. Food and Drug Administration (FDA) and by European Medicines Agency (EMA): Olaparib, rucaparib, niraparib and talazoparib. Olaparib was the first PARP inhibitor to be approved, in 2014, as maintenance therapy for platinum-sensitive relapsed advanced OC with germline *BRCA 1/2* mutations, and these results were confirmed by the SOLO-2 trial [20]. More recently, the SOLO-1 trial demonstrated the benefit of using olaparib as maintenance therapy after chemotherapy in *BRCA 1/2* mutated patients [21]. Other PARP inhibitors have been approved for the maintenance treatment of recurrent, epithelial ovarian, fallopian tube, or primary peritoneal cancer irrespective of the *BRCA* status, such as rucaparib and niraparib.

For BC, olaparib was first approved in 2018 for treating patients with germline *BRCA1/2* mutations HER2-negative metastatic BC. The benefit of this approach was demonstrated in the phase III OlympiAD study that assessed olaparib monotherapy versus chemotherapy [18]. In the same setting of BC disease, talazoparib was studied in the EMBRACA trial and also demonstrated a survival benefit [19]. In the early-stage BC setting, the OlympiA trial reported a significant benefit for patients with a high risk of recurrence and germline *BRCA1/2* mutations treated with adjuvant olaparib following the completion of standard therapy [22]. *BRCA* testing was previously used in BC only to predict the risk of future cancers and guide surgical therapies. However, with the recent advent of PARP inhibitors, it may be reasonable to consider genetic testing more widely than past criteria have considered [23].

Many studies about the efficacy of chemotherapy have reported high sensitivity in patients with *BRCA* mutations. Chemotherapy agents that have a mechanism of action for DNA damage, such as anthracyclines and cyclophosphamide, or DNA replication, such as platinum compounds, may explain the high effectiveness in *BRCA* mutated patients [7].

### 2.1. The Founder Effect

*BRCA1/2* is the most frequent alteration associated with HBOC and can affect all ethnic groups. The frequency of *BRCA1/2* alteration in the general population has been estimated at one in 400–500 [4]. More than 2000 different mutations have been identified since the discovery of these genes. The most frequent mutations are small insertion/deletion frameshift, non-synonymous truncation and splice-site disruption leading to entirely non-functional BRCA proteins [24]. 

The incidence of mutations in high-risk families varies widely among different populations. However, in particular ethnic groups that are or were geographically or culturally isolated, specific mutations show a high frequency. This is often called the “founder effect” [4].

Founder mutations of *BRCA1/2* were described in specific populations. A typical example of a founder effect is the Ashkenazi Jewish population (ancestors from Eastern and Central Europe) with a prevalence of 1:40 [25]. Founder variants in *BRCA1/2* have also been reported in several additional populations, including individuals of African, Amish, and Icelandic ancestry [26]. Identifying founder mutations is informative and valuable for developing cancer gene screening panels, which help to analyse genetic susceptibility profiles [5].

### 2.2. Portuguese Founder Mutations

With the overseas exploration from the 15th century and after many foreign invasions occurred in the Iberia peninsula, connections with people of different cultures and genetic backgrounds were established. However, several recurrent mutations which integrate the *BRCA1/2* spectrum mutations were found in Portugal.

The c.156_157insAlu *BRCA2* mutation is the most described in different studies (Table 1). It was first reported by Teugels, et al. in a 46-years-old Portuguese woman with BC living in Belgium with diagnosed HBOC. It was demonstrated that an Alu insertion in *BRCA2* exon 3 was responsible for an in-frame deletion at the mRNA level, not detectable using the conventional mutation screening techniques [6].

Based on this finding, Machado, et al. evaluated the ancestral origin and population spread of c.156_157insAlu *BRCA2* mutation, using a three-step PCR procedure for the molecular and phenotypic characterisation [27]. Those authors found a regional founder effect for this rearrangement in HBOC families mostly originated from central/southern Portugal and suggested that the founder event occurred 2400–2600 years ago [27]. Peixoto, et al. reported the mutation in 14 of 208 families from northern/southern Portugal, and its contribution to more than one-fourth of pathogenic *BRCA1/BRCA2* mutations in HBOC families originated from this part of the country [28].

In 2010, the same authors screened c.156_157insAlu *BRCA2* rearrangement in a total of 5443 suspected HBOC families from Europe, North and South America and Asia (149 from Portugal and 5294 from other countries than Portugal). Three main conclusions were published: (1) Contrary to the Machado, et al. results, after an extensive haplotype analysis in 11 informative families, the age of this founder mutation was estimated to be 558 ± 215 years; (2) This mutation was only detected in families with Portuguese ancestry living in France and USA; (3) It accounted for the majority of the *BRCA2* mutations and about one-third of all pathogenic germline mutations in Portuguese HBOC families [30].

Other studies were conducted regarding the mutational prevalence and clinical characteristics of c.156_157insAlu *BRCA2*. In a multidisciplinary HBOC programme at Instituto Português de Oncologia (IPO) de Lisboa, a group of 3566 suspected individuals were tested exclusively for *BRCA1/2*, in which 386 patients were diagnosed with a *BRCA1/2* mutation. Of these, 22.8 % had the c.156_157insAlu *BRCA2* rearrangement. An analysis conducted on individuals with Madeira Island ancestry revealed two patients with the rearrangement from 19 tested. When comparing the pattern of *BRCA1/2* mutation in this subgroup, a lower prevalence of the Portuguese *BRCA2* founder variant was observed (10.5 vs. 23.4%) [31].

According to the different studies mentioned, the Portuguese geographical distribution of c.156_157insAlu *BRCA2* mutation is shown in Figure 1.

A small study conducted in IPO de Coimbra pointed to the clinical characteristics of 78 patients from central Portugal with the c.156_157insAlu *BRCA2* mutation followed in this oncology centre. Two patients (7.4%) had OC, four patients (14.8%) had bilateral BC at diagnosis, and nine patients (33.3%) had second cancers some years later (breast, prostate, colorectal and skin). Related to BC, “luminal B-like” was the most frequent subtype (55.2%). The prevalence of “Luminal A-like”, “Luminal B-HER2+”, and TNBC were 24.1%, 6.9% and 13.8%, respectively. None of the patients was a stage IV at diagnosis, but 30% had metastatic progression during the surveillance time [32].

Other possible founder mutation genes in Portugal are being discussed. In addition to c.156_157insAlu in *BRCA2*, Peixoto, et al. also reported two other mutations in the *BRCA1* gene (c.2037delinsCC and c.3331_3334del). Those three represent about 50% of all pathogenic mutations in Portuguese HBOC families. A preliminary haplotype study of c.2037delinsCC mutation revealed that three families with this mutation share a common ancestor [30]. The c.3331_3334del mutation was reported in different populations, including Spain, Africa, Canada, Brazil and Colombia. Tuazon, et al. suggested that this mutation originated in the Iberia peninsula and was later introduced in South America. Moreover, both mutations reported in *BRCA1* shared a common haplotype, indicating the possibility of being founder mutations in the Portuguese population [33].

## 3. *BRCA* Gene Testing and Screening

Germline mutations in the *BRCA1/2* genes are associated with a higher risk of BC and OC [34]. In healthy women, access to *BRCA* status could allow prophylactic procedures, reducing the risk or even improving survival. Prophylactic bilateral mastectomy provides a 90% to 95% risk reduction, but the available data do not demonstrate improved mortality. Bilateral salpingo-oophorectomy, on the contrary, translates to an improvement in survival [35,36]. 

Nevertheless, it has obvious fertility and menopausal impact, underscoring the complexity of this issue [36]. For instance, follow-up is also individualised: annual breast magnetic resonance imaging (MRI) screening should be commenced from the age of 25 with the addition of annual mammography from the age of 30 [37]. Conversely, identifying a pathogenic variant in a woman diagnosed with BC may influence treatment and prognosis [38,39].

Genetic testing should only be performed after adequate information is yielded by a trained health professional who can explain the implications of the results. A health care expert can perform this counselling, ideally a genetic counsellor or clinical geneticist. Due to the lack of genetic expertise in many centres, it is unlikely to provide all eligible women formal and complete genetic counselling on an equitable basis [40,41]. Because of the high costs associated with genetic analyses, *BRCA1/2*- testing has been limited to BC patients with an a priori high risk of being carriers of a pathogenic variant [42].

Acknowledging one’s risk for testing positive for a genetic mutation has clinical and personal implications; thus, accurate risk assessment through guidelines or risk assessment models is critical.

The American Society of Clinical Oncology (ASCO), The National Comprehensive Cancer Network (NCCN) and the European Society of Medical Oncology (ESMO) all have guidelines for *BRCA* testing based on these risk factors. In fact, more than fifteen guidelines on *BRCA* testing are available in Europe, and more than thirty guidelines are available worldwide [43]. Despite the high number of recommendations regarding the diagnosis and management of this population, most guidelines do not represent international consensus, and practice usually follows national/international guidelines on a country-by-country basis. Still, the ESMO and NCCN guidelines are internationally recognised and should be mentioned. 

Most guidelines recommend genetic counselling and testing for patients at heightened risk of a *BRCA* pathogenic variant. Specific criteria help to calculate this increased risk, such as young age at diagnosis, TNBC or a family history of BC, OC, pancreatic or prostatic cancer [44]. The definition of young age differs slightly between the guidelines: it is characterised in some guidelines as being under 50 years of age at diagnosis and in others under 45 years of age. The ESMO guidelines are more conservative regarding the population of patients with a diagnosis of TNBC: they consider an indication for testing only in patients under the age of 60 years. In other guidelines, such as those of the NCCN, the testing criteria in the TNBC population are independent of age. Additional criteria include the diagnosis of multiple primary BC (synchronous or metachronous), lobular BC with personal or family history of diffuse gastric cancer and male BC. Regarding the family history, close blood relatives with a diagnosis of cancer (BC, OC, pancreatic cancer, metastatic or high-grade prostatic cancer) grant a higher risk for the presence of a pathogenic variant. It is also recommended to be tested at any age to aid in treatment decisions of PARP inhibitors in the metastatic setting, as previously discussed [41,45].

The guidelines also mention indications for testing individuals who otherwise do not meet the criteria but have a probability > 5% of a pathogenic variant based on risk assessment models—e.g., Tyrer-Cuzick, BRCAPro, CanRisk [41]. These genetic risk assessment models have been used to predict an individual’s likelihood of possessing a *BRCA* gene mutation. The Tyrer-Cuzick model, also named the IBIS tool, provides a risk score that estimates the probability of developing BC. It is calculated using an assortment of risk factors, including personal health history and family history. The score is usually expressed as a percentage [45]. The BRCAPRO model is a risk assessment tool that utilises Bayes Mendel analysis and ultimately determines those at a higher risk of developing BC, OC and other cancer types. The BRCAPRO model is an accurate model for determining the probability of carrying a genetic mutation [46]. Similar to the Tyrer-Cuzick model, it also incorporates personal and family history information. Finally, the CanRisk tool is a web interface to the Breast and Ovarian Analysis of Disease Incidence and Carrier Estimation Algorithm (BOADICEA). This model also incorporates family history, personal lifestyle, hormonal and reproductive risk and mammographic density, and it was described by Lee, et al. [47]. These models have allowed users to precisely tailor calculations for patients and families. However, although a woman’s risk may be accurately estimated, these predictions do not allow one to say which woman will develop BC. On the other hand, it has been suggested that if genetic testing were performed only on BC patients meeting the NCCN guidelines testing criteria, nearly half of patients with a germline mutation associated with HBOC syndrome would not be identified [48]. 

Regarding the combined use of clinical criteria and risk model tools, Peixoto et al. characterised the mutational spectrum of germline *BRCA1*/*BRCA2* mutations in 1050 Portuguese HBOC families, of which 524 were fully screened [30]. BRCAPRO mutation probability was also retrospectively calculated for the group of probands that had been selected based on clinical criteria. Inherited cancer predisposition could be related to *BRCA1* or *BRCA2* mutations in 21.4% of the 524 probands, a proportion that increases to 28.9% of the families with an a priori BRCAPRO mutation probability >10%. Seven additional pathogenic mutations were detected in the 526 families with BRCAPRO mutation probability <10% that were screened only for the two most frequent mutations (c.156_157insAlu and c.3331_3334del). Of the 119 families with pathogenic mutations, 33.6% had the *BRCA2* c.156_157insAlu rearrangement, 12.6% the *BRCA1* c.3331_3334del mutation and 5.9% the *BRCA1* c.2037delinsCC mutation [30].

Identifying specific founder variants allows a more efficient and cost-saving mutational screening approach. It enables the oncologist to make more specific choices, simplifying the clinical process of genetic testing on high-risk family members [30]. Likewise, a frequent founder mutation allows a more precise assessment of mutation-specific cumulative cancer incidence, facilitating the identification of genetic and environmental risk modifiers. It is also important to mention that in the guidelines, the relevance of founder mutations is reinforced by the suggestion that testing for three founder mutations of *BRCA1/2* could be offered to individuals with one grandparent identified as Ashkenazi Jewish ancestry, irrespective of cancer history in the family [41]. This strategy has been confirmed to be cost-effective in the Ashkenazi Jewish populations due to the mutational pattern (three mutations constitute the majority of harmful mutations in the Jewish population) and the high frequency of these mutations in this population [49].

Expanding population-based testing to other populations has obstacles that we must overcome. We know that two mutations in *BRCA1* (c.2037delinsCC and c.3331_3334del) and one in *BRCA2* (c.156_157insAlu) together represent about 50% of all harmful mutations found in affected Portuguese families [30]. We hence suggest that all suspected families with Portuguese ancestry living around the world should include, but not be limited to, the analysis of *BRCA2* c.156_157insAlu and *BRCA1* c.3331_3334del mutations. This is relevant because it requires a specific PCR reaction, not carried out by most commercial laboratories [50].

## 4. Future Perspectives 

In the science community, fundamental research is making efforts to characterise the genetic features of *BRCA*. Recently, Silva, et al. investigated the tissue-specific stem/progenitor cells that represent the cells of origin of *BRCA2*-associated tumours. In this article, the authors reported the generation of the first lineage of induced pluripotent stem cell (iPSC) from a female donor harbouring the *BRCA2* c.156_157insAlu mutation, which is a valuable tool for studying the origin of *BRCA2*-associated cancer in its earliest phases [51]. Pinheiro, et al. identified and described the genomic breakpoints of two duplications that occurred in tandem and in a direct direction in the *BRCA1* and *BRCA2* genes. They considered that these mutations are the genetic defect underlying HBOC syndrome in these families [52].

Cancer risk genes other than *BRCA1* and *BRCA2* are also under investigation. Many studies have revealed non-*BRCA* germline pathogenic mutations in high-risk individuals, evidencing the importance of NGS analysis in high-risk HBOC genes. Supporting this data, Salgueiro, et al. reported two cases with co-occurrence of pathogenic mutations in breast and ovarian high-risk cancer genes: One proband had co-occurrence of pathogenic variants in *BRCA1* and *RAD51C* genes and the other proband in *PALB2* and *CHEK2* [53]. The same authors showed in a different study with 211 patients who met NCCN guidelines for HBOC genetic testing, a rate of cancer-predisposing pathogenic/likely pathogenic (P/LP) variants in non-*BRCA1/BRCA2* genes of 6.2%, which is consistent with those reported in the literature [52].

The occurrence of double heterozygosity (DH) in *BRCA1* and *BRCA2* genes and double mutation (DM) in *BRCA1* or *BRCA2* were also described. Among 645 patients, two probands with DH in the *BRCA1/2* genes not previously described together were found. Furthermore, three probands with DM in the *BRCA2* gene were reported in further unrelated patients and likely pointed to a founder effect. Although rare, the detection of these alterations could have important clinical implications for managing patients and risk assessment in the family members of mutated patients [54].

## 5. Conclusions

Hereditary tumours are syndromes characterised by the expression of multiple types of cancer in succession, including hereditary BC or OC. Hence, we discussed the importance of determining the genetic predisposition of HBOC families in specific populations, pointing out the Portuguese reality. Since the first case reported of *BRCA2* c.156_157insAlu and later described as a Portuguese founder mutation, many efforts were performed to gain insight into the ancestral origin and its population spread. While mutational heterogeneity was reported, other frequent mutations were also found in subsequent investigations. Many models have been designed to statistically predict *BRCA1/2* based on personal and family histories concerning the pathological and clinical significance. Although the thresholds observed may not be enough to determine the high probability of strictly incorporating pathogenic variants, these models should be used to predict *BRCA* mutation carriers. In addition, cancer risk genes other than *BRCA1* and *BRCA2* cannot be overlooked, which may contribute to HBOC susceptibility. 

Taking advantage of knowing HBOC predisposition, it is essential to adopt protective measures for relatives, offer preventive measures to avoid or diagnose earlier other cancers and discuss treatment options.

## Figures and Tables

**Figure 1 cancers-14-04717-f001:**
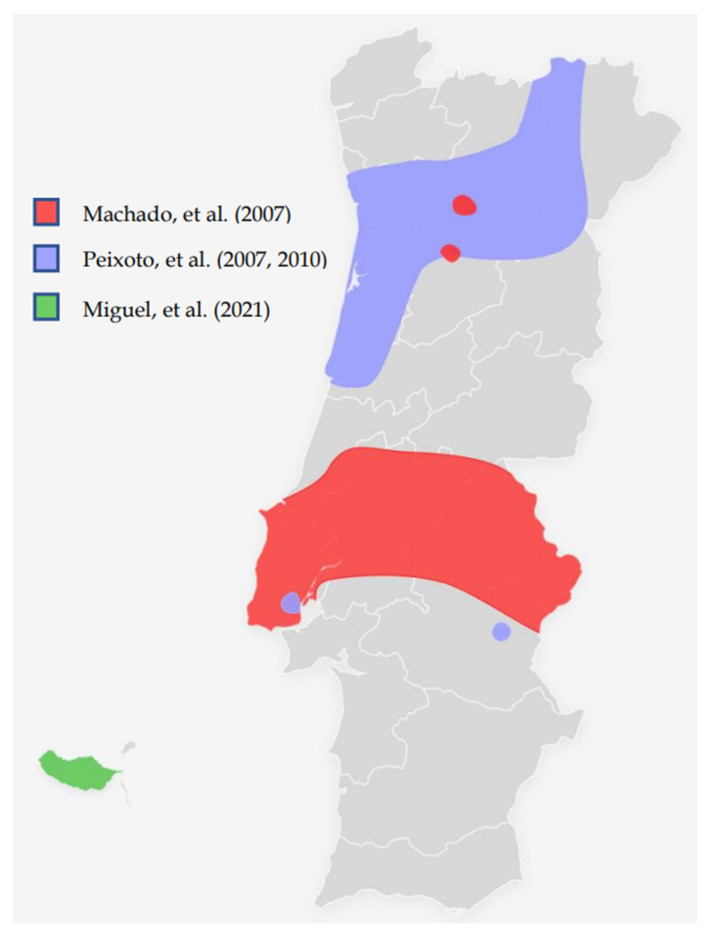
Map of Portugal showing the known geographical origin of the families with the c.156_157insAlu BRCA2 germline mutations according to the different studies [27,28,31].

**Table 1 cancers-14-04717-t001:** Relevant studies about the prevalence of the founder mutations in Portugal.

Studies	Aim	Number of Patients/Families	Founder Mutation Investigated	Conclusions
Machado, et al. (2007) [27]	Molecular and phenotypic characterisation of a large insertion in exon 3 of *BRCA2.*	210 patients from Central/southern Portugal	c.156_157insAlu	c.156_157insAlu is a founder mutation of Portuguese origin and is the most frequent *BRCA2* rearrangement
Peixoto, et al. (2008) [28]	To evaluate the contribution of the c.156_157insAlu *BRCA2* mutation to inherited predisposition to BC and OC in families originating mostly from northern/central Portugal.	210 families from Northern/central Portugal	c.156_157insAlu	This rearrangement is responsible for more than half of all pathogenic *BRCA2* mutations and about one-fourth of all pathogenic variants in HBOC families
Peixoto, et al. (2010) [29]	To gain insight into the ancestral origin and population spread of the c.156_157insAlu *BRCA2* mutation.	5443 families (149 from Portugal and 5294 from other countries than Portugal)	c.156_157insAlu	c.156_157insAlu *BRCA2* rearrangement is a Portuguese founder mutation that originated about 558 ± 215 years ago, accounting for the majority of the *BRCA2* mutations and about one-third of all pathogenic germline mutations in Portuguese HBOC families.
Peixoto, et al. (2015) [30]	To describe the mutational spectrum and evaluate the impact of founder mutations in the genetic testing criteria and strategy for molecular testing of HBOC families of Portuguese ancestry.	1050 families (524 fully screened for *BRCA1*/*BRCA2* mutations)	c.156_157insAlu Other possible founder mutations pointed out: c.3331_3334del c.2037delinsCC	Of the 119 families with pathogenic mutations, 40 (33.6%) had the *BRCA2* c.156_157insAlu rearrangement, 15 (12.6%) the *BRCA1* c.3331_3334del mutation and 7 (5.9%) the BRCA1 c.2037delinsCC mutation. The c.2037delinsCC mutation has not been described in other populations.
Miguel, et al. (2021) [31]	To evaluate the Hereditary Breast/Ovarian Cancer (HBOC) families with Madeira ancestry enrolled in the HBOC programme occurring in Instituto Português de Oncologia de Lisboa	3566 patients (19 from Madeira Island, 3547 from other parts of Portugal)	c.156_157insAlu	*BRCA1/2* detection rates were 27.9% and 10.5% for Madeira and the whole group, respectively. In all patients detected with BRCA1/2 mutations, 22.8 % had the c.156_157insAlu *BRCA2* rearrangement.

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
