# Peer review of "Mutation Patterns in Portuguese Families with Hereditary Breast and Ovarian Cancer Syndrome"

_cancers, 2022, doi:10.3390/cancers14194717_

Round 1

Reviewer 1 Report (Previous Reviewer 1)

I am satisfied with the edits made. Thank you.

Author Response

Dear reviewers,

At first, we would like to thank all the pertinent comments and suggestions that contributed to an accurate and precise approach to this subject.

In the previous version submitted, we attempted to overcome the off-topic information according to the pertinent annotations made. One of the significant concerns pointed out by the editor was the excessive information related to BRCAness phenotypes and to other cancers not related to the main subject of this article. For this reason, in order to condense and direct the information, it was necessary to focus on BRCA mutation features and to suppress information less related to the founder variants in the Portuguese population. Although recognising the importance of epigenetic alterations in the field, we added the mention in line 73 that alterations of the BRCA1/2 genes may also occur through mechanisms other than germline mutations, for example, somatic mutations or epigenetic silencing in sporadic OC or BC. 

In addition, we carry out a more careful linguistic review.

We hope that the aforementioned changes meet the reviewers’ expectations.

We are available for any clarification.

Kind regards,

The authors

Reviewer 2 Report (Previous Reviewer 2)

Although described in the point-by-point letter, much of the revisions required are not well integrated in the revised version. For example, the possible impact of epigenetics in such type of cancer.

Author Response

Dear reviewers,

At first, we would like to thank all the pertinent comments and suggestions that contributed to an accurate and precise approach to this subject.

In the previous version submitted, we attempted to overcome the off-topic information according to the pertinent annotations made. One of the significant concerns pointed out by the editor was the excessive information related to BRCAness phenotypes and to other cancers not related to the main subject of this article. For this reason, in order to condense and direct the information, it was necessary to focus on BRCA mutation features and to suppress information less related to the founder variants in the Portuguese population. Although recognising the importance of epigenetic alterations in the field, we added the mention in line 73 that alterations of the BRCA1/2 genes may also occur through mechanisms other than germline mutations, for example, somatic mutations or epigenetic silencing in sporadic OC or BC. 

In addition, we carry out a more careful linguistic review.

We hope that the aforementioned changes meet the reviewers’ expectations.

We are available for any clarification.

Kind regards,

The authors

This manuscript is a resubmission of an earlier submission. The following is a list of the peer review reports and author responses from that submission.

Round 1

Reviewer 1 Report

Summary:

 This is a review aimed at describing mutational rearrangements of Portuguese founder mutations although the authors also review indications for testing and recommendations for carriers overall. 

The review is well organized, it is easy to read and provides some interesting data. This is a relevant topic focused on founder mutations in the Portuguese population with the goal of delivering more directed and cost effective genetic testing strategies. The manuscript can benefit from expanding more on the topic of the mutation patters in Portuguese families and from shortening the sections that are not directly related to this topic. For example, section 3 occupies almost 2 pages in this manuscript and is not directly related to the main topic.   

Major comments:

-       1. Consider expanding on the therapeutic implications of identifying a BRCA mutations on those with BRCA associated breast cancer. It would be interesting to include how results of OlympiA, EMBRACA and Olympia trials change our perspective on the reasons and indications for testing. Should all patients who meet criteria for these trials be tested? If not, why. Can comment on whether PARPi are available in USA and Europe in the metastatic and adjuvant setting. This is a significant change in the reasons why we perform genetic testing. On page 2 line 73, an opportunity to comment on this topic is lost here. There is only mention to the reason why genetic testing has been historically ordered and availability of NGS. Doesn’t comment on whether there are therapeutic implications for many patients that do not undergo NGS.

-         2. Consider commenting on the availability of genetic counselors, and whether it is feasible to have all patients undergo genetic counseling pre-genetic testing.

-         3. Page 3, line 112: I don’t believe this is an accurate statement, there is no definitive evidence supporting worse BC survival in carriers and the authors didn’t include a reference to support this. Studies have been contradictory in this field. Consider expanding more on this topic and providing appropriate references or simply deleting it.

-          4. Page 6, line 216: not clear in this paragraph if the authors are referring that BSO and bilateral mastectomy are associated with survival or only one of the interventions. The paper cited included both interventions but there is only strong survival data related to BSO. Please clarify in the text.

-          5. Please review English carefully

Minor comments:

-          Please make sure to italicize BRCA1, BRCA2 or BRCA1/2 throughout all the manuscript. Shouldn’t italicize when refers to the protein.

Author Response

Dear reviewer,

"please see the attachment."

Reviewer 2 Report

The manuscript is adequate for the journal. The scientific message may be of interest to the community involved in the study of BRCA1/2 in the pathogenesis of cancer. Surprisingly, the authors did not quote and discuss the revant work of Dr. Vietri in the field (see and quote the following refs: Genes (Basel). 2022 Feb 9;13(2):321. doi: 10.3390/genes13020321.; Int J Oncol. 2021 Nov;59(5):98. doi: 10.3892/ijo.2021.5278.; Med Oncol. 2021 Jan 23;38(2):13. doi: 10.1007/s12032-021-01454-5.; Genes (Basel). 2020 Dec 3;11(12):1451. doi: 10.3390/genes11121451.; Eur J Med Genet. 2020 Jun;63(6):103883. doi: 10.1016/j.ejmg.2020.103883. ; Clin Chem Lab Med. 2012 Dec;50(12):2171-80. doi: 10.1515/cclm-2012-0154.). Epigenetic programming of such involvement would be also discussed.

Moreover, the flow text needs a major revision of English grammar.

Author Response

(The authors gave the same response as above.)
